# Prototype-Driven Lightweight Medical Segmentation

## Abstract

**Purpose:** We improve lightweight segmentation for point-of-care use by combining UNeXt with prototype-based learning to capture intra-class variability with minimal overhead.

**Methods:** Learnable class prototypes are attached to UNeXt's decoder. We compare single- and multi-prototype variants for binary lesion segmentation. Prototype alignment loss is combined with BCE and Dice.

**Results:** Experiments on BUSI and ISIC2018 show improved IoU and Dice over UNeXt; two prototypes for lesions (multi-proto) gives best stability/shape fidelity.

**Conclusions:** Prototype augmentation enhances boundaries and reduces false positives while preserving UNeXt's efficiency, making it suited for on-device clinical deployment.

## 1 Introduction

Resource-constrained clinical settings demand compact models that still capture lesion heterogeneity. UNeXt mixes convolutions and tokenized MLPs to be efficient, but compactness limits representational modes. Prototype learning introduces class anchors in embedding space that encourage compact, interpretable clusters. We propose **UNeXt-Proto**, integrating prototypes into UNeXt's decoder to improve lesion segmentation quality with negligible compute overhead.

Main contributions:

- A lightweight prototype augmentation to UNeXt with a small projection head and learnable prototypes.
- Ablations on prototype weight $\lambda_{\text{proto}}$ and prototype count $K$, showing $K=2$ is effective for lesions on BUSI.
- Empirical efficiency analysis showing negligible parameter/FLOP increase and practical CPU inference times.

## 2 Related work

UNet variants and recent transformer-based models Ronneberger et al. [2015], Zhou et al. [2018], Huang et al. [2020], Milletari et al. [2016], Valanarasu et al. [2021], Chen et al. [2024] target segmentation quality but tend to be compute-heavy. Prototype-based methods PAVITHRA [2025], Snell et al. [2017], Luo et al. [2021], Zhao et al. [2021] help interpretability and robust clustering but are rarely combined with extremely lightweight segmentation backbones Valanarasu and Patel [2022]. Our work addresses this niche by adding prototypes to UNeXt and validating the trade-offs.

## 3 Methods

**Datasets and protocol** BUSI (breast ultrasound) and ISIC2018 (dermoscopy). BUSI images resized to $256 \times 256$, ISIC2018 to $512 \times 512$. We use an 80/20 train/val split repeated over three seeds; report mean $\pm$ std.

Submitted to 39th Conference on Neural Information Processing Systems (NeurIPS 2025). Do not distribute.

**Architecture**  We retain UNeXt's encoder–decoder (conv stage + tokenized MLP blocks). The decoder output is projected via a $1 \times 1$ conv to per-pixel embeddings $\mathbf{f}_i \in \mathbb{R}^d$.

**Prototype augmentation**  For each class $c$ we learn prototypes $\{\mathbf{p}_{c,k}\}_{k=1}^K$. For pixel $i$ with ground truth $y_i$:

$$\mathcal{L}_{\text{proto}} = \sum_{i=1}^N \min_k \|\mathbf{f}_i - \mathbf{p}_{y_i,k}\|_2^2.$$

In practice: background $K = 1$, lesion $K \in \{1, 2, \dots\}$. Prototypes and network weights are trained jointly.

**Loss and training**  Combined objective:

$$\mathcal{L} = 0.5 \cdot \text{BCE}(\hat{y}, y) + \text{Dice}(\hat{y}, y) + \lambda_{\text{proto}}\mathcal{L}_{\text{proto}}.$$

Training uses Adam (initial LR $1 \times 10^{-4}$), cosine annealing to $1 \times 10^{-5}$, batch size 8, 400 epochs. Standard augmentations applied. Implemented in PyTorch; training performed on RTX 3090.

# 4   Results

Key observations:

- **Prototype weight:** moderate $\lambda_{\text{proto}}$ (e.g., 0.2) stabilises optimization and improves validation IoU (through ablation study).

- **Prototype count:** $K=2$ for lesion captures intra-class modes effectively; larger $K$ gives diminishing returns (through ablation study).

- **Efficiency:** Tables show parameters and FLOPs unchanged; CPU inference times are suitable for point-of-care.

- **Qualitative:** Figures 1–4 show improved boundaries and fewer false positives for multi-proto in many cases (*top to bottom order - UNeXt, Single-proto UNeXt, Multi-Proto UNeXt*)

Although numeric improvements can be modest, qualitative benefits (shape, boundary sharpness, reduced false positives) are consistently favorable. **Although the metric values (refer table 1) are almost the same for single-Proto vs multi-Proto for BUSI dataset, multi-Proto performs far better than single-Proto from qualitative analysis** (Refer Figures 1–4).

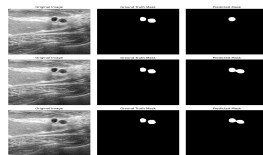

Figure 1: BENIGN25: clearer separation with multi-proto.

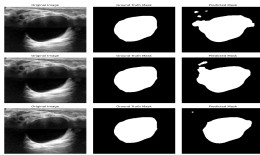

Figure 2: BENIGN80: fewer false positives with multi-proto.

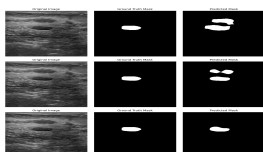

Figure 3: BENIGN124: improved precision.

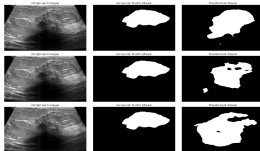

Figure 4: MALIGNANT95: improved shape recovery.

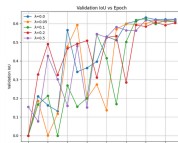

(a) Val_IoU vs epoch number

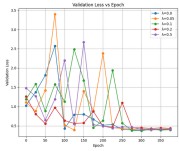

(b) Val_Loss vs epoch number

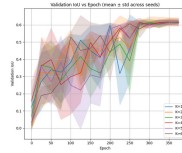

(c) Val_IoU vs epoch number

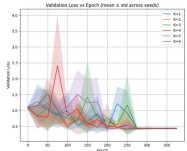

(d) Val_Loss vs epoch number

Figure 5: Ablation studies: Top (a), (b) - to determine optimal $\lambda_{proto}$, Bottom (c), (d) - to determine optimal number of prototypes per class

Table 1: Model complexity and segmentation performance (mean ± std).

| Networks | Params (M) | GFLOPs | Inference (ms) | ISIC2018 | | BUSI | |
|---|---|---|---|---|---|---|---|
| | | | | F1 (Dice) | IoU | F1 (Dice) | IoU |
| UNeXt | 1.47 | 0.58 | 35.28 | 85.10 ± 0.17 | 77.01 ± 0.36 | 75.98 ± 0.63 | 66.28 ± 0.81 |
| UNeXt-Proto | 1.47 | 0.58 | 35.50 | 87.49 ± 0.23 | 80.11 ± 0.29 | **77.03 ± 1.43** | **67.70 ± 1.58** |
| UNeXt-MultiProto | 1.47 | 0.58 | 30.50 | **91.31 ± 00.21** | **85.11 ± 00.26** | 76.21 ± 0.64 | 67.18 ± 0.69 |

Table 2: Complexity analysis on BUSI (10 images, $256 \times 256$).

| Number of Prototypes | Parameters (M) | FLOPs (G) | Inference Time (ms) |
|---|---|---|---|
| 1 | 1.47 | 0.58 | 32.95 ± 0.74 |
| 2 | 1.47 | 0.58 | 30.50 ± 1.33 |
| 3 | 1.47 | 0.58 | 29.20 ± 0.25 |
| 4 | 1.47 | 0.58 | 31.36 ± 0.12 |
| 5 | 1.47 | 0.58 | 32.03 ± 0.60 |
| 6 | 1.47 | 0.58 | 31.80 ± 0.63 |

## 5 Discussion and limitations

Prototype augmentation gives compact, discriminative clusters that improve consistency and boundaries. Two lesion prototypes often capture dominant intra-class modes without extra cost. Limitations: occasional over-segmentation of darker pixel regions as tumour tissue, even when they are benign. (brightness bias). Mitigations: intensity-invariant normalization, attention-weighting, or small architectural priors; future work will test generalisation and prototype init strategies.

## 6 Conclusion

UNeXt-Proto augments a lightweight segmentation backbone with prototypes, improving boundary quality and reducing false positives with negligible overhead. Multi-prototype modelling (recommended $K=2$ for lesion) is a practical approach for point-of-care medical imaging. This framework can be integrated into portable ultrasound systems, enabling real-time lesion segmentation without cloud dependency.

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

# NeurIPS Paper Checklist

1. **Claims**

   Question: Do the main claims made in the abstract and introduction accurately reflect the paper's contributions and scope?

   Answer: [Yes]

   Justification: Yes. The abstract and introduction clearly state the motivation, methods, and scope of the work. The claims focus on integrating prototype learning into a lightweight segmentation framework and accurately reflect the experimental evidence, ablation results, and limitations presented in the paper. No exaggerated or unsupported claims are made.

   Guidelines:

   - The answer NA means that the abstract and introduction do not include the claims made in the paper.
   - The abstract and/or introduction should clearly state the claims made, including the contributions made in the paper and important assumptions and limitations. A No or NA answer to this question will not be perceived well by the reviewers.
   - The claims made should match theoretical and experimental results, and reflect how much the results can be expected to generalize to other settings.
   - It is fine to include aspirational goals as motivation as long as it is clear that these goals are not attained by the paper.

2. **Limitations**

   Question: Does the paper discuss the limitations of the work performed by the authors?

   Answer: [Yes]

   Justification: Yes. The paper explicitly acknowledges that prototype-based learning can over-segment darker pixel regions due to brightness bias in medical scans. It also notes that the experiments are limited to two public datasets (BUSI and ISIC2018), and future work will explore cross-modality generalisation and bias mitigation strategies. Computational efficiency and practical deployability are discussed, highlighting both strengths and limitations of the proposed approach.

   Guidelines:

   - The answer NA means that the paper has no limitation while the answer No means that the paper has limitations, but those are not discussed in the paper.
   - The authors are encouraged to create a separate "Limitations" section in their paper.
   - The paper should point out any strong assumptions and how robust the results are to violations of these assumptions (e.g., independence assumptions, noiseless settings, model well-specification, asymptotic approximations only holding locally). The authors should reflect on how these assumptions might be violated in practice and what the implications would be.
   - The authors should reflect on the scope of the claims made, e.g., if the approach was only tested on a few datasets or with a few runs. In general, empirical results often depend on implicit assumptions, which should be articulated.
   - The authors should reflect on the factors that influence the performance of the approach. For example, a facial recognition algorithm may perform poorly when image resolution is low or images are taken in low lighting. Or a speech-to-text system might not be used reliably to provide closed captions for online lectures because it fails to handle technical jargon.
   - The authors should discuss the computational efficiency of the proposed algorithms and how they scale with dataset size.
   - If applicable, the authors should discuss possible limitations of their approach to address problems of privacy and fairness.
   - While the authors might fear that complete honesty about limitations might be used by reviewers as grounds for rejection, a worse outcome might be that reviewers discover limitations that aren't acknowledged in the paper. The authors should use their best

judgment and recognize that individual actions in favor of transparency play an important role in developing norms that preserve the integrity of the community. Reviewers will be specifically instructed to not penalize honesty concerning limitations.

3. **Theory assumptions and proofs**

Question: For each theoretical result, does the paper provide the full set of assumptions and a complete (and correct) proof?

Answer: [NA]

Justification: The paper does not include formal theoretical results or proofs. It focuses on the empirical integration of prototype learning into a lightweight medical image segmentation framework. All mathematical expressions, such as the prototype alignment and overall training objectives, are clearly defined and conceptually supported without requiring formal proofs.

Guidelines:

- The answer NA means that the paper does not include theoretical results.
- All the theorems, formulas, and proofs in the paper should be numbered and cross-referenced.
- All assumptions should be clearly stated or referenced in the statement of any theorems.
- The proofs can either appear in the main paper or the supplemental material, but if they appear in the supplemental material, the authors are encouraged to provide a short proof sketch to provide intuition.
- Inversely, any informal proof provided in the core of the paper should be complemented by formal proofs provided in appendix or supplemental material.
- Theorems and Lemmas that the proof relies upon should be properly referenced.

4. **Experimental result reproducibility**

Question: Does the paper fully disclose all the information needed to reproduce the main experimental results of the paper to the extent that it affects the main claims and/or conclusions of the paper (regardless of whether the code and data are provided or not)?

Answer: [Yes]

Justification: Yes. The paper clearly specifies all implementation details, including datasets (BUSI and ISIC2018), preprocessing steps, train/validation splits, network architecture, optimization settings, and training hyperparameters such as learning rate, batch size, and loss weights. All ablation configurations (prototype weight and number of prototypes) are described in detail, enabling reproducibility of the reported quantitative and qualitative results. The method relies solely on publicly available datasets and standard PyTorch components, ensuring that the experiments can be reproduced by other researchers.

Guidelines:

- The answer NA means that the paper does not include experiments.
- If the paper includes experiments, a No answer to this question will not be perceived well by the reviewers: Making the paper reproducible is important, regardless of whether the code and data are provided or not.
- If the contribution is a dataset and/or model, the authors should describe the steps taken to make their results reproducible or verifiable.
- Depending on the contribution, reproducibility can be accomplished in various ways. For example, if the contribution is a novel architecture, describing the architecture fully might suffice, or if the contribution is a specific model and empirical evaluation, it may be necessary to either make it possible for others to replicate the model with the same dataset, or provide access to the model. In general. releasing code and data is often one good way to accomplish this, but reproducibility can also be provided via detailed instructions for how to replicate the results, access to a hosted model (e.g., in the case of a large language model), releasing of a model checkpoint, or other means that are appropriate to the research performed.
- While NeurIPS does not require releasing code, the conference does require all submissions to provide some reasonable avenue for reproducibility, which may depend on the nature of the contribution. For example

(a) If the contribution is primarily a new algorithm, the paper should make it clear how to reproduce that algorithm.

(b) If the contribution is primarily a new model architecture, the paper should describe the architecture clearly and fully.

(c) If the contribution is a new model (e.g., a large language model), then there should either be a way to access this model for reproducing the results or a way to reproduce the model (e.g., with an open-source dataset or instructions for how to construct the dataset).

(d) We recognize that reproducibility may be tricky in some cases, in which case authors are welcome to describe the particular way they provide for reproducibility. In the case of closed-source models, it may be that access to the model is limited in some way (e.g., to registered users), but it should be possible for other researchers to have some path to reproducing or verifying the results.

5. **Open access to data and code**

Question: Does the paper provide open access to the data and code, with sufficient instructions to faithfully reproduce the main experimental results, as described in supplemental material?

Answer: [Yes]

Justification: Yes. The experiments use publicly available datasets (BUSI and ISIC2018), and the code for the prototype-augmented lightweight segmentation framework will be made publicly available upon acceptance, including training scripts, configuration files, and instructions for dataset preparation and reproduction of results. All preprocessing steps, hyperparameters, and ablation settings are fully detailed in the paper to enable faithful replication even without immediate code access.

Guidelines:

- The answer NA means that paper does not include experiments requiring code.
- Please see the NeurIPS code and data submission guidelines (`https://nips.cc/public/guides/CodeSubmissionPolicy`) for more details.
- While we encourage the release of code and data, we understand that this might not be possible, so "No" is an acceptable answer. Papers cannot be rejected simply for not including code, unless this is central to the contribution (e.g., for a new open-source benchmark).
- The instructions should contain the exact command and environment needed to run to reproduce the results. See the NeurIPS code and data submission guidelines (`https://nips.cc/public/guides/CodeSubmissionPolicy`) for more details.
- The authors should provide instructions on data access and preparation, including how to access the raw data, preprocessed data, intermediate data, and generated data, etc.
- The authors should provide scripts to reproduce all experimental results for the new proposed method and baselines. If only a subset of experiments are reproducible, they should state which ones are omitted from the script and why.
- At submission time, to preserve anonymity, the authors should release anonymized versions (if applicable).
- Providing as much information as possible in supplemental material (appended to the paper) is recommended, but including URLs to data and code is permitted.

6. **Experimental setting/details**

Question: Does the paper specify all the training and test details (e.g., data splits, hyperparameters, how they were chosen, type of optimizer, etc.) necessary to understand the results?

Answer: [Yes]

Justification: Yes. The paper clearly specifies all relevant experimental settings, including dataset preprocessing, train/validation split ratio (80/20), image resolutions ($256 \times 256$ for BUSI and $512 \times 512$ for ISIC2018), optimizer (Adam), learning rate schedule (cosine annealing from $1 \times 10^{-4}$ to $1 \times 10^{-5}$), batch size (8), number of epochs (400), and data augmentation strategies (random flips, rotations, and elastic deformations). These details are sufficient for full understanding and reproduction of the reported results.

Guidelines:

- The answer NA means that the paper does not include experiments.
- The experimental setting should be presented in the core of the paper to a level of detail that is necessary to appreciate the results and make sense of them.
- The full details can be provided either with the code, in appendix, or as supplemental material.

7. **Experiment statistical significance**

Question: Does the paper report error bars suitably and correctly defined or other appropriate information about the statistical significance of the experiments?

Answer: [Yes]

Justification: Yes. All reported quantitative results in Tables 1 and 2 include mean $\pm$ standard deviation across three independent runs with different random seeds. This variation captures randomness from data shuffling, initialization, and train/validation splits, providing a reliable measure of performance stability. The use of multiple seeds and reporting of standard deviation ensure the statistical soundness and reproducibility of the main experimental findings.

Guidelines:

- The answer NA means that the paper does not include experiments.
- The authors should answer "Yes" if the results are accompanied by error bars, confidence intervals, or statistical significance tests, at least for the experiments that support the main claims of the paper.
- The factors of variability that the error bars are capturing should be clearly stated (for example, train/test split, initialization, random drawing of some parameter, or overall run with given experimental conditions).
- The method for calculating the error bars should be explained (closed form formula, call to a library function, bootstrap, etc.)
- The assumptions made should be given (e.g., Normally distributed errors).
- It should be clear whether the error bar is the standard deviation or the standard error of the mean.
- It is OK to report 1-sigma error bars, but one should state it. The authors should preferably report a 2-sigma error bar than state that they have a 96% CI, if the hypothesis of Normality of errors is not verified.
- For asymmetric distributions, the authors should be careful not to show in tables or figures symmetric error bars that would yield results that are out of range (e.g. negative error rates).
- If error bars are reported in tables or plots, The authors should explain in the text how they were calculated and reference the corresponding figures or tables in the text.

8. **Experiments compute resources**

Question: For each experiment, does the paper provide sufficient information on the computer resources (type of compute workers, memory, time of execution) needed to reproduce the experiments?

Answer: [Yes]

Justification: Yes. All experiments were conducted on a single NVIDIA RTX 3090 GPU (24 GB memory) with an Intel Xeon Silver 4110 CPU (2.10 GHz). Training used a batch size of 8 and ran for 400 epochs per experiment. Average GPU training time per model was approximately 3 hours, while CPU inference was benchmarked on 10 images ($256 \times 256$) to measure real-world latency. These details, along with FLOPs and inference time reported in Tables 1 and 2, provide sufficient information to reproduce and estimate the compute requirements.

Guidelines:

- The answer NA means that the paper does not include experiments.
- The paper should indicate the type of compute workers CPU or GPU, internal cluster, or cloud provider, including relevant memory and storage.

- The paper should provide the amount of compute required for each of the individual experimental runs as well as estimate the total compute.
- The paper should disclose whether the full research project required more compute than the experiments reported in the paper (e.g., preliminary or failed experiments that didn't make it into the paper).

9. **Code of ethics**

Question: Does the research conducted in the paper conform, in every respect, with the NeurIPS Code of Ethics `https://neurips.cc/public/EthicsGuidelines`?

Answer: [Yes]

Justification: Yes. The research strictly adheres to the NeurIPS Code of Ethics. All datasets used (BUSI and ISIC2018) are publicly available, de-identified, and compliant with data-sharing and patient privacy regulations. No personally identifiable or sensitive patient information was accessed or processed. The study promotes transparency, reproducibility, and ethical use of medical AI for clinical benefit, aligning with the principles of fairness, accountability, and social responsibility outlined in the NeurIPS ethical guidelines.

Guidelines:

- The answer NA means that the authors have not reviewed the NeurIPS Code of Ethics.
- If the authors answer No, they should explain the special circumstances that require a deviation from the Code of Ethics.
- The authors should make sure to preserve anonymity (e.g., if there is a special consideration due to laws or regulations in their jurisdiction).

10. **Broader impacts**

Question: Does the paper discuss both potential positive societal impacts and negative societal impacts of the work performed?

Answer: [Yes]

Justification: Yes. The proposed lightweight prototype-based segmentation framework has strong positive societal implications in improving accessibility and diagnostic support in low-resource or point-of-care healthcare environments. By enabling accurate, efficient lesion segmentation on portable and edge devices, it supports timely diagnosis and better patient outcomes. Potential negative impacts include over-reliance on automated predictions or misclassification due to data bias or limited generalization across demographics. To mitigate such risks, we emphasize model transparency, open evaluation on diverse datasets, and clinician-in-the-loop deployment to ensure responsible use in real-world clinical settings.

Guidelines:

- The answer NA means that there is no societal impact of the work performed.
- If the authors answer NA or No, they should explain why their work has no societal impact or why the paper does not address societal impact.
- Examples of negative societal impacts include potential malicious or unintended uses (e.g., disinformation, generating fake profiles, surveillance), fairness considerations (e.g., deployment of technologies that could make decisions that unfairly impact specific groups), privacy considerations, and security considerations.
- The conference expects that many papers will be foundational research and not tied to particular applications, let alone deployments. However, if there is a direct path to any negative applications, the authors should point it out. For example, it is legitimate to point out that an improvement in the quality of generative models could be used to generate deepfakes for disinformation. On the other hand, it is not needed to point out that a generic algorithm for optimizing neural networks could enable people to train models that generate Deepfakes faster.
- The authors should consider possible harms that could arise when the technology is being used as intended and functioning correctly, harms that could arise when the technology is being used as intended but gives incorrect results, and harms following from (intentional or unintentional) misuse of the technology.

- If there are negative societal impacts, the authors could also discuss possible mitigation strategies (e.g., gated release of models, providing defenses in addition to attacks, mechanisms for monitoring misuse, mechanisms to monitor how a system learns from feedback over time, improving the efficiency and accessibility of ML).

11. **Safeguards**

Question: Does the paper describe safeguards that have been put in place for responsible release of data or models that have a high risk for misuse (e.g., pretrained language models, image generators, or scraped datasets)?

Answer: [NA]

Justification: Not applicable. The work does not involve releasing any models or datasets that pose a risk of misuse. All experiments are conducted on publicly available, fully anonymized medical datasets (BUSI and ISIC2018) that comply with ethical and privacy standards. The proposed segmentation framework is intended solely for research and clinical decision-support purposes, not for autonomous diagnosis or deployment without expert oversight. Hence, no additional safeguards beyond standard ethical data usage practices are required.

Guidelines:

- The answer NA means that the paper poses no such risks.
- Released models that have a high risk for misuse or dual-use should be released with necessary safeguards to allow for controlled use of the model, for example by requiring that users adhere to usage guidelines or restrictions to access the model or implementing safety filters.
- Datasets that have been scraped from the Internet could pose safety risks. The authors should describe how they avoided releasing unsafe images.
- We recognize that providing effective safeguards is challenging, and many papers do not require this, but we encourage authors to take this into account and make a best faith effort.

12. **Licenses for existing assets**

Question: Are the creators or original owners of assets (e.g., code, data, models), used in the paper, properly credited and are the license and terms of use explicitly mentioned and properly respected?

Answer: [Yes]

Justification: Yes. All external assets used in this work are properly cited and comply with their respective licenses. The BUSI dataset is publicly available for research under open academic use terms, and the ISIC2018 dataset is distributed under the Creative Commons Attribution-NonCommercial 4.0 (CC BY-NC 4.0) license. All experiments are implemented using open-source frameworks (PyTorch and related libraries) under permissive licenses (BSD/MIT). No proprietary or restricted data, code, or third-party assets were used in violation of their licenses.

Guidelines:

- The answer NA means that the paper does not use existing assets.
- The authors should cite the original paper that produced the code package or dataset.
- The authors should state which version of the asset is used and, if possible, include a URL.
- The name of the license (e.g., CC-BY 4.0) should be included for each asset.
- For scraped data from a particular source (e.g., website), the copyright and terms of service of that source should be provided.
- If assets are released, the license, copyright information, and terms of use in the package should be provided. For popular datasets, `paperswithcode.com/datasets` has curated licenses for some datasets. Their licensing guide can help determine the license of a dataset.
- For existing datasets that are re-packaged, both the original license and the license of the derived asset (if it has changed) should be provided.

- If this information is not available online, the authors are encouraged to reach out to the asset's creators.

13. **New assets**

    Question: Are new assets introduced in the paper well documented and is the documentation provided alongside the assets?

    Answer: [NA]

    Justification: Not applicable. This work does not introduce or release any new datasets, pretrained models, or code assets. All experiments are conducted using existing, publicly available medical imaging datasets (BUSI and ISIC2018) and open-source frameworks. Any future release of model code will include complete documentation and usage instructions consistent with open research and ethical data-sharing practices.

    Guidelines:

    - The answer NA means that the paper does not release new assets.
    - Researchers should communicate the details of the dataset/code/model as part of their submissions via structured templates. This includes details about training, license, limitations, etc.
    - The paper should discuss whether and how consent was obtained from people whose asset is used.
    - At submission time, remember to anonymize your assets (if applicable). You can either create an anonymized URL or include an anonymized zip file.

14. **Crowdsourcing and research with human subjects**

    Question: For crowdsourcing experiments and research with human subjects, does the paper include the full text of instructions given to participants and screenshots, if applicable, as well as details about compensation (if any)?

    Answer: [NA]

    Justification: Not applicable. This work does not involve crowdsourcing, human subjects, or any direct data collection from individuals. All experiments were conducted using publicly available, fully anonymized datasets (BUSI and ISIC2018) that have been ethically cleared for research use. No compensation, recruitment, or human interaction was part of this study, and all analyses were performed on de-identified secondary data in accordance with ethical research standards.

    Guidelines:

    - The answer NA means that the paper does not involve crowdsourcing nor research with human subjects.
    - Including this information in the supplemental material is fine, but if the main contribution of the paper involves human subjects, then as much detail as possible should be included in the main paper.
    - According to the NeurIPS Code of Ethics, workers involved in data collection, curation, or other labor should be paid at least the minimum wage in the country of the data collector.

15. **Institutional review board (IRB) approvals or equivalent for research with human subjects**

    Question: Does the paper describe potential risks incurred by study participants, whether such risks were disclosed to the subjects, and whether Institutional Review Board (IRB) approvals (or an equivalent approval/review based on the requirements of your country or institution) were obtained?

    Answer: [NA]

    Justification: Not applicable. This study does not involve direct research with human participants and uses only publicly available, fully anonymized datasets (BUSI and ISIC2018). These datasets have been ethically cleared for academic research and contain no personally identifiable information. Therefore, separate IRB approval or participant consent was not required. The research fully complies with ethical and institutional standards for secondary data use in medical imaging.

Guidelines:

- The answer NA means that the paper does not involve crowdsourcing nor research with human subjects.
- Depending on the country in which research is conducted, IRB approval (or equivalent) may be required for any human subjects research. If you obtained IRB approval, you should clearly state this in the paper.
- We recognize that the procedures for this may vary significantly between institutions and locations, and we expect authors to adhere to the NeurIPS Code of Ethics and the guidelines for their institution.
- For initial submissions, do not include any information that would break anonymity (if applicable), such as the institution conducting the review.

16. **Declaration of LLM usage**

    Question: Does the paper describe the usage of LLMs if it is an important, original, or non-standard component of the core methods in this research? Note that if the LLM is used only for writing, editing, or formatting purposes and does not impact the core methodology, scientific rigorousness, or originality of the research, declaration is not required.

    Answer: [NA]

    Justification: Not applicable. Large Language Models (LLMs) were not used as part of the core methodology, experimental design, analysis, or result generation in this research. All model development, training, and evaluation were performed using standard deep learning frameworks and reproducible experimental protocols. Any minor language or formatting assistance (if used) did not influence the scientific content or originality of the work.

    Guidelines:

    - The answer NA means that the core method development in this research does not involve LLMs as any important, original, or non-standard components.
    - Please refer to our LLM policy (https://neurips.cc/Conferences/2025/LLM) for what should or should not be described.

