# OpenReview forum: "Prototype-Driven Lightweight Medical Segmentation"
_EurIPS.cc/2025/Workshop/MedEurIPS — EurIPS 2025 Workshop MedEurIPS Submission_

### Official Review · Reviewer_q1jZ · 2025-10-27
**Prototype-Driven Lightweight Medical Segmentation**

**Rating:** 6
**Confidence:** 4

**Review:**

Summary:
The paper presents a method for lesion segmentation that uses prototype learning to improve lesion discrimination. The method employs a UNeXt backbone with an additional projection layer to learn the prototypes. Experimental results demonstrate that this approach achieves higher performance than the vanilla architecture without a significant computational overhead.

Strengths:
- Lesion segmentation is highly relevant for the medical AI community.
- The paper is clearly written and easy to follow.
- The results indicate consistent improvements the baseline. Also, the improvement with multiple prototypes are significant.

Weaknesses:
- The technical novelty is limited. The method evaluates an established prototype learning method with a new backbone.
- The figures use too much space that could be better used for more methodological details.
- The references and citations are in the wrong format

Overall assessment:
The paper proposes an interesting approach for lesion segmentation in medical imaging. The technical contribution is relatively minor, but the observed performance gains are substantial.

---

### Official Review · Reviewer_14q3 · 2025-10-28
**Review comments**

**Rating:** 3
**Confidence:** 5

**Review:**

The paper presents a UNeXt variant with prototype learning for medical image segmentation, showing performance gains over the vanilla UNeXt on two datasets.

The technical contribution is limited. The authors fail to provide clear motivation or justification for choosing the UNeXt architecture specifically for this proposed method.

The academic writing quality is poor. Specifically, the paper needs a major revision with a more detailed discussion of related work and a more thorough description of the proposed method.

The evaluation is incomplete. Comparisons with prototype-based segmentation methods and other state-of-the-art medical image segmentation methods should be included.

The figure resolution is too low, making visualization difficult. Additionally, the plot legends are too small and need to be enlarged for clarity.

---

### Decision · Program_Chairs · 2025-10-31

**Decision:**

Reject

**Comment:**

Both reviewers find the topic relevant and the empirical results consistent but agree that the technical novelty is limited, calling for deeper analysis, stronger baselines, and improvements in writing and figure quality.